# Expression of Peptidyl Arginine Deiminase 2 Is Closely Associated with Recurrence in Patients with Hepatocellular Carcinoma

**DOI:** 10.3390/diagnostics13040659

**Published:** 2023-02-10

**Authors:** Sunho Uhm, Yoon Ah Cho, Ji-Young Choe, Ji Won Park, Min-Jeong Kim, Won-Ho Han, Junyong Lee, Jung Woo Lee, Dong Woo Shin, Jae Seung Soh, Hyun Lim, Ho Suk Kang, Sung-Hoon Moon, Sung-Eun Kim

**Affiliations:** 1Department of Internal Medicine, Hallym University Sacred Heart Hospital, Hallym University College of Medicine, Anyang 14068, Republic of Korea; 2Department of Pathology, Hallym University Sacred Heart Hospital, Hallym University College of Medicine, Anyang 14068, Republic of Korea; 3Anatomic Pathology Reference Lab, Seegene Medical Foundation, Seoul 04805, Republic of Korea; 4Institute for Liver and Digestive Diseases, Hallym University, Chuncheon 24252, Republic of Korea; 5Department of Radiology, Hallym University Sacred Heart Hospital, Anyang 14068, Republic of Korea; 6Department of Anesthesiology and Pain Medicine, Seoul Medical Center, Seoul 02053, Republic of Korea; 7Department of Surgery, Hallym University Sacred Heart Hospital, Hallym University College of Medicine, Anyang 14068, Republic of Korea

**Keywords:** peptidyl arginine deiminase, hepatocellular carcinoma, recurrence, survival, hepatic resection

## Abstract

Peptidyl arginine deiminases (PAD) enzymes have been investigated in various cancers. Recently, PAD enzyme, in particular PAD2, has been further implicated in cancers. Although the expression of PAD2 was significantly higher in hepatocellular carcinoma (HCC) tissue, its diagnostic or prognostic role of PAD2 in HCC patients is unknown. This study investigated whether the expression of PAD2 affects recurrence and survival in HCC patients who underwent hepatic resection. One hundred and twenty-two HCC patients after hepatic resection were enrolled. The median follow-up was 41 months (range 1–213 months) in enrolled patients. To investigate an association between PAD2 expression level and the clinical characteristics of enrolled patients, the recurrence of HCC following surgical resection and survival of the patients were examined. Ninety-eight cases (80.3%) of HCC demonstrated a higher expression of PAD2. The expression of PAD2 was correlated with age, hepatitis B virus positivity, hypertension, and higher alpha-fetoprotein level. There was no association between PAD2 expression and sex, diabetes mellitus, Child–Pugh class, major portal vein invasion, HCC size or number. The recurrence rates in patients with lower PAD2 expression were higher than those with higher PAD2 expression. The cumulative survival rates of patients with higher PAD2 expression were better than those of patients with lower PAD2 expression, but it was not statistically significant. In conclusion, PAD2 expression is closely associated with recurrence of HCC patients following surgical resection.

## 1. Introduction

Liver cancer is the sixth most common cancer and the fourth leading cause of cancer-related death worldwide in 2018 [1]. Among liver cancer, hepatocellular cellular carcinoma (HCC) accounts for approximately 90% of liver cancers; over than 90% of HCC patients had underlying chronic liver disease (CLD) and especially cirrhosis of any etiology [2,3]. Treatment of HCC has several issues compared to other cancers. First, about 90% of HCC patients have poor underlying liver function because they generally have CLD such as chronic hepatitis B, chronic hepatitis C, alcoholic hepatitis, nonalcoholic fatty liver disease/steatohepatitis or cirrhosis. Second, surgical treatment for HCC such as resection and liver transplantation are applied to 30–40% of patients in Western counties and a smaller proportion of patients in Asia. Third, the risk of recurrence is relatively higher than other cancers, and it continues even after 5 years or more after treatment [2,3]. Fourth, although recently systemic chemotherapies, including immune checkpoint inhibitors, tyrosine kinase inhibitors and monoclonical antibodies have tried to cure HCC, it could not show satisfiable results in HCC patients [2,3,4]. Therefore, the discovery of target proteins for HCC is still necessary to develop new anticancer drugs, and it is enhancing the effects of existing anticancer drugs.

Citrullination is one of the post-translational modifications that converses the positive charges of arginine residues to citrulline residues via calcium-dependent peptidylarginine deiminase (PAD) representing five different isoforms, type 1–4 and type 6, which have distinct substrate and tissue specificity [5,6,7,8]. Citrulline is a nonstandard amino acid and no citrulline tRNA exists; citrulline residues in proteins are only the product of citrullination by PAD, leading to alterations in protein structure and function [6,7]. The PAD has roles in regulating cellular processes such as cell proliferation, differentiation, survival, and death in the nervous and immune systems. Although PAD normally exists as an inactive form, it becomes active and induces the citrullination of cellular proteins when the intracellular calcium balance breaks. Interestingly, several studies have shown that PAD expression is associated with cancer proliferation, metastasis, and drug resistance in various cancers [7,9]. Among them, PAD2 is the most widely distributed type in mammalian tissues, especially in skeletal muscle, brain, secretory gland, dermis, spleen, and hematopoietic cells [8,10].

PAD2 has been implicated in various cancers such as gastrointestinal cancers, HCC, breast cancer, lung cancer, thyroid cancer, and skin tumor [11,12]. Guo et al. showed that the expression of PAD2 in tissue and blood samples of HCC, gastric cancer, and colon cancer was significantly higher than in those of healthy subjects [11]. Compared to the study of Guo et al., Funayama et al. demonstrated that the expression of PAD2 in colon cancer tissue was decreased and citrullination was also suppressed in colon cancer tissue [13]. In addition, low PAD2 expression in colorectal cancer tissues was associated with poor outcome in these patients [14]. Although PAD2 expression in HCC has been reported previously, the association between PAD2 expression and prognosis of HCC patients has not been investigated. Therefore, this study aimed to investigate whether the expression of PAD2 in HCC tissues affects the recurrence and survival in HCC patients underwent surgical resection.

## 2. Materials and Methods

### 2.1. Patients and Tissue Microarray

A total of 122 HCC patients that underwent surgical resection were enrolled from January 2011 to December 2015 at Hallym University Sacred Heart Hospital in South Korea. The enrolled patients were followed up for a median period of 41 months (range 1–213 months) after surgical resection. Recurrence of HCC and survival of the enrolled patients were checked using medical records at the last follow-up date. HCC patients who satisfied all of the inclusion criteria were included: (1) no previous treatment for HCC before surgical treatment; (2) histologic confirmation of HCC after surgical resection; (3) complete tumor resection with negative margins; (4) a single lesion smaller than 8 cm; and (5) up to three lesions, each smaller than 3 cm. Enrolled patients were followed up once a month for the first months and then every 3 months thereafter. Complete blood count (CBC), liver function tests, prothrombin time (PT), and serum alpha-fetoprotein (AFP) were checked routinely at the postoperative follow-up. In addition, computed tomography (CT) or magnetic resonance imaging (MRI) was performed every 3 months for surveillance of HCC recurrence. If HCC recurred during follow-up, each patient received optimal treatment for HCC such as transarterial chemoembolizaion, radiofrequency ablation, systemic chemotherapy, etc. Patient’s survival was determined from death certificates or outpatient clinic/phone follow-up. Informed consent was waived because this study was retrospective and analyzed anonymous clinical data. This study was approved by the ethics committee of Hallym University Sacred Heart Hospital. The study was conducted with the approval of the Institutional Review Board of Hallym University Sacred Heart Hospital (HALLYM 2021-05-009-002).

### 2.2. Immunohistochemical Staining of PAD2 in a Human HCC Tissue Microarray

Tissue microarrays (TMAs) were constructed in which a 2 mm-sized tissue core was perforated and re-embedded from the labeled lesions. Each tissue core was arrayed in duplicate to preserve HCC tissue. Then, TMA blocks were cut into 4 µm thick slices and dried for 1 h at 60 °C. TMA slices were deparaffinized with xylene and hydrated in a graded ethanol series to water. Antigen retrieval was performed using a citrate-based antigen unmasking solution (H-3300; Vector laboratories, Burlingame, CA, USA) in the pressure cooker for 10 min. Slides were washed in PBS. Endogenous peroxidase activity was blocked with endogenous blocking solution (SP-6000, Vector laboratories, Newark, CA, USA) for 10 min. Samples were blocked for a further 30 min with BSA (SP-5050, Vector laboratories) blocking buffer and incubated with rabbit polyclonal antibody-PADI2/PAD2 (1:1000; ab231911, Abcam. Cambridge, UK) overnight at 4 °C. After primary antibody incubation, the samples were washed in PBS with 0.5% Triton X-100. Slides were incubated at room temperature with a biotinylated goat anti-rabbit IgG secondary antibody (BA-1000, Vector laboratories) for 1 h and with an avidin–biotin complex (ABC) reagent (PK-6100, Vector laboratories) for 30 min. Samples were washed in PBST before and after application of the ABC reagent. Chromogenic development was achieved using the DAB Peroxidase Substrate kit (SK-4100, Vector laboratories). After a brief rinse in tap water, the sections were counterstained with Harris hematoxylin and cover slipped using a synthetic mounting medium.

### 2.3. Evaluation of Immunostaining

PAD2 stained slides were used to examine the proportion of tumor cells that were positive for PAD2 in the tumor cell nuclei. For each spot, areas of the most intense and/or predominant staining patterns were scored. Their intensity of PAD2 immunohistochemical staining (IHC) was usually positively correlated with the proportion of positive tumor cells. To analyze the TMA, slides were measured semiquantitative using a four-graded system based on the intensity and distribution similar to previous study [15]: graded 0, no staining (0); graded 1, weak (+), graded 2; medium (++), graded 3; strong (+++). Degrees of all TMAs IHC were investigated by light microscopy (BX51; Olympus, UK). Based on the criteria, all TMAs were classified and analyzed in correlation with the clinical data: (1) low PAD2 expression (less than 50%, none or grade 1) and (2) high PAD2 expression (over 50%, grade 2 or grade 3). All tissue samples were confirmed by two experienced pathologists (Cho YA, Choe JY). Two independent pathologists determined PAD2 expression levels, and both pathologists re-examined specimens with discrepant scores to determine a consensus score.

### 2.4. Statistical Analysis 

Continuous variables are expressed as the mean ± SD or median (interquartile range). Categorical variables are presented as number of patients (%). For statistical significance, Student’s *t*-test and χ2 test were conducted for comparisons of variables with normal distribution between groups. In addition, a Mann–Whitney test was conducted for comparisons of variables with non-normal distribution between groups. The cumulative recurrence of HCC and survival were evaluated by the Kaplan–Meier method, and differences were determined by the log-rank test. Univariate and multivariate analysis were carried out to identify independent predictors for HCC recurrence and patient’s survival using a Cox regression hazard model. Deaths from only hepatic origin were included for overall survival analysis. A value of *p* < 0.05 was regarded as statistically significant. All analyses were performed using SPSS 27.0 software (SPSS, Inc., Chicago, IL, USA).

## 3. Results

### 3.1. Baseline Characteristics of Enrolled Patients

Table 1 demonstrated baseline characteristics according to PAD2 expression in enrolled patients. One hundred and four patients (85.2%) were male. Sixty-seven patients (47.5%) had liver cirrhosis. The majority of the enrolled patients (91.8%, 112/122) had well-reserved liver function with Child–Pugh class A. A total of 88 patients had HCC less than 5 cm in diameter, and 34 had HCC more than 5 cm in diameter. The mean tumor number was 1.24 ± 0.59. The baseline characteristics of the 122 patients are listed in Table 1.

### 3.2. Differences According to PAD2 Expression

Of the 122 HCC samples, PAD2 was stained in 114 (93.4%), but only eight HCC patients were not stained. Of 113 positively stained samples, 16 were grade 1, 62 were grade 2, and 36 HCC TMAs showed grade 3 expression (Figure 1). The proportion of patients with high PAD2 staining (grade 2 or grade 3) was 80.3% (98/110) compared to 19.7% (24/122) with low PAD2 staining (0 or grade 1). To evaluate the clinical significance of PAD2 in HCC, the association between the expression level of PAD2 and the clinical features of patients with HCC was analyzed. Patients with low PAD2 expression showed younger age, hepatitis B surface antigen (HBsAg) positivity, hypertension, and higher AFP level (*p* < 0.05). However, the expression of PAD2 did not show significant differences with sex, diabetes mellitus (DM), liver cirrhosis, underlying liver function, tumor size, tumor number, CBC, PT, and vascular invasion, as shown in Table 1.

### 3.3. Association of PAD2 Expression with Survival in HCC Patients Underwent Surgical Resection

To determine the effect of PAD2 expression on overall survival in HCC patients who underwent surgical resection, we conducted Kaplan–Meier analysis to compare the survival between patients with high PAD2 expression and low PAD2 expression. The cumulative survival rates of patients with high PAD2 expression did not show any significant difference from those of patients with low PAD2 expression (*p* = 0.311) (Figure 2).

### 3.4. Univariate and Multivariate Analysis of Overall Survival in HCC Patients Underwent Surgical Resection

Although the overall survival of HCC patients with low PAD2 expression did not show any significant difference compared to patients with high PAD2 expression, tumor size was revealed to be only one risk factor for the overall survival in enrolled patients (Table 2).

### 3.5. Association of PAD2 Expression with Recurrence in HCC Patients Underwent Surgical Resection

Kaplan–Meier analysis was used to compare the cumulative recurrence rate and confirm the association between the expression level of PAD2 (low vs. high) and the recurrence of hepatocellular carcinoma. The 1-, 3- and 5-year cumulative recurrence rates of high PrP^C^ expression patients was significantly lower than those of low PrP^C^ expression patients (19.1%, 38.3% and 46.9% vs. 50%, 66.7% and 75.0%, respectively; *p* = 0.011) (Figure 3).

### 3.6. Univariate and Multivariate Analysis of Recurrence in HCC Patients Underwent Surgical Reaction 

Univariate and multivariate Cox analyses were performed to determine related factors for the HCC recurrence in patients who underwent surgical resection. On univariate analysis, we found that male sex, larger tumor size (≥5 cm), higher AFP level (≥400 ng/mL), and low PAD2 expression were potential candidates for multivariate analysis of survival (*p* < 0.05 on univariate analysis). After multivariate analysis, male sex (OR 0.500, 95% CI 0.253–0.986, *p* = 0.045), larger tumors (OR 2.522, 95% CI 1.430–4.554, *p* = 0.002) and low PAD2 expression (OR 2.007, 95% CI 1.010–4.075, *p* = 0.048) were independent risk factors for postoperative recurrence (Table 3).

## 4. Discussion

The role of PAD is essentially to induce citrullination, which affects cell differentiation, apoptosis, inflammation, tissue destruction, autoimmunity, and cancer [6,9,16]. Increasing evidence from several studies supported the important role of PAD2 in the development and progression of various cancers [17]. Several studies demonstrated that PAD2 is abnormally regulated in gastrointestinal carcinogenesis [11,13,14]. Guo et al. reported that the expression of PAD2 was increased in human HCC tissue; however, they did not show clinical significance of PAD2 in HCC [11]. Abdeen et al. found that PAD is expressed only in the liver tissue of patients with chronic hepatitis and hepatic fibrosis, but not in normal liver, and the degree of hepatic fibrosis and inflammation correlates with the intensity of PAD IHC [18]. Another study of Abdeen et al. reported that the serum anti-modified citrullinated vimentin level was significantly increased in hepatic fibrosis compared to patients with no hepatic fibrosis [19]. Circulating levels of citrullinated and matrix metalloproteinases (MMP)-degraded vimentin (VICM) were increased in mouse liver tissues of hepatic fibrosis and in human liver tissues of early hepatic fibrosis associated with hepatitis c virus (HCV) and nonalcoholic fatty liver disease [20]. In a rat model of CCl4-induced hepatic fibrosis, the VICM level was decreased with the level of PAD after treatment of a pan-PAD inhibitor. Therefore, PAD inhibitor led to a significant decrease in VCIM level and might be a potential drug for hepatic fibrosis [21]. In addition, our previous study reported that PAD2 and citrullinated proteins, especially citrullinated glial fibrillary acidic protein (GFAP), are increased in hepatic fibrosis [22]. Therefore, we hypothesized that PAD2 would be a factor influencing the prognosis of HCC patients.

In this study, we first demonstrated that the expression of PAD was closely associated with recurrence and survival in HCC patients who underwent surgical resection. Although 93.9% of HCC showed PAD2 expression, there were different recurrence rates according to the degree of PAD2 expression in HCC tissue. There was a statistical difference between PAD2 expression and age/HBsAg positivity/HTN/AFP level. It is possible that the expression of PAD2 may increase due to the higher probability of intracellular calcium homeostasis being broken with age, but there has been no report on the relationship between age and PAD2 expression in CLD patients. Abdeen et al. demonstrated that the expression of PAD was increased in human liver biopsy samples from chronic hepatitis and hepatic fibrosis but not from normal liver, and they also showed that the degree of hepatic fibrosis and hepatic inflammation is associated with the IHC degree of PAD immunochemical staining. Although the number of enrolled patients of this study was relatively small (46 HCV infection, 20 hepatitis B virus (HBV) infection, 8 nonalcoholic steatohepatitis, 8 autoimmune hepatitis, 10 others, and 8 healthy subjects) [18], their results suggested that chronic insults for liver might be associated with PAD2 expression. Our study showed that the positivity of HBsAg was higher in patients with low PAD2 expression. Interestingly, HCC patients with low PAD2 expression had a higher recurrence rate of HCC. Although no study has yet elucidated the association between HBV infection and PAD expression, it is considered necessary to confirm the effect of HBV infection, which is the main cause of HCC, on the expression of PAD as an additional study. In this study, there was no association between PAD2 expression and sex, liver cirrhosis, underlying liver function, tumor size, tumor number, or platelet count. As a result, our data showed that the degree of PAD2 expression may not show a direct correlation with underlying liver function, but further investigation is warranted. PAD2 expression tended to be decreased and protein citrullination was suppressed in colon cancer patients compared with matched healthy controls [13]. In addition, Cantariño et al. showed that colon cancer patients with high PAD2 expression in the tumor or the adjacent mucosa have the best prognosis and, conversely, the downregulation of PAD2 correlates with poor survival [14]. It can be speculated that in addition to a tumor-intrinsic function, PAD2 might modulate the microenvironment in a way that its absence favors tumor progression. PAD2 induced the proliferation of HCT-116 human colon cancer cells transfected with FLAG epitope-tagged PAD2 expression vectors but does not affect the apoptosis of these cells [13]. HCT-116 cells overexpressing PAD2 demonstrated that an increased proportion of cells is found in G1 phase, while a decreased proportion is found in S phase [13]. In addition, PAD2-mediated citrullination of β-catenin inhibits the proliferation of SW480 and HCT116 colon cancer cells by blocking the wingless (Wnt)/β-catenin pathway [23]. The suppression of PAD2 via anti-PAD2 siRNA treatment significantly increases the proliferation and migration of MKN-45 human gastric cancer cells and attenuates the apoptosis of these cells, but the decreased effects are observed in Bel-7402 liver cancer cells [11]. Therefore, further study is needed to confirm the role of PAD in HCC unlike other cancers such as colon cancer, gastric cancer, etc.

Disease-free survival rates were not different in HCC patients between low and high PAD2 expression; however, the recurrence rate of HCC after surgical resection was significantly higher in HCC patients with low PAD2 expression (*p* = 0.010). The present study revealed that larger tumor size (≥5 cm) and low PAD2 expression in HCC tissue were independent risk factors for postoperative recurrence in HCC patients. Because the number of subjects in this study was relatively small, more patients are needed to confirm the role of PAD2 in HCC. Furthermore, additional studies on the association between changes in citrullinated protein via PAD2 and HCC are needed. This study suggests that the expression of PAD2 in human HCC tissue might be a prognostic indicator in HCC patients.

In 2006, Luo et al. reported the first bioactive PAD inhibitor when they described the synthesis and characterization of F-amidine that is bioavailable and has been used to confirm that PAD activity can inhibit gene expression [24]. Cl-amidine, like F-amidine, was generated by the Thompson group, it has been the most widely used PAD inhibitor, and studies indicate that it is bioavailable and can inhibit PAD4 activity [25,26]. Interestingly, Cl-amidine was cytotoxic to cancer cell lines (MCF-7, HL-60, and HT29), but it demonstrated minimal cytotoxic effects on noncancerous cell lines (NIH 3T3 and HL-60 granulocytes) [26]. Cui et al. also reported that that Cl-amidine induced p53 and inhibits cell growth in a p53-dependent manner in HCT116 colon cancer cell line [27]. Furthermore, Cl-amidine has been investigated on numerous disease models and shows promise for the treatment and/or prevention of many diseases [7]. However, there is no study using PAD inhibitors in HCC yet and considering the shortage of effective anticancer drugs for HCC, studies using PAD inhibitors in HCC are expected.

Our study has some limitations. First, because the number of patients was relatively small and HCC patients who underwent surgical resection were enrolled, it is not sufficient to evaluate the association of PAD2 expression according to each stage of HCC and the role of PAD2 chemoresistance in HCC. In addition, the number of patients with low PAD2 expression was only 24. Second, there was an inherent bias in the design because it was a retrospective manner. Therefore, our results need to be confirmed in large prospective studies. Third, we did not experimentally investigate the pathophysiologic mechanism of PAD2 in HCC. Therefore, investigating the expression of PAD2 in blood or other body fluid is also warranted in HCC patients. Fourth, the role of citrullinated protein should be investigated because citrullinated protein via PAD2 as a final product is the most important target protein in HCC.

## 5. Conclusions

We found that PAD2 expression is closely associated with recurrence in HCC patients who underwent surgical resection. Therefore, the expression level of PAD2 in HCC tissues might be a potential prognostic marker after curative surgery. The discovery of biomarkers predicting outcomes such as recurrence and survival of HCC and target proteins for the cure of HCC is still a major issue in clinical setting. Further studies are required to identify protein substrates of PAD2 that control HCC development and progression and to elucidate how citrullinated proteins influence HCC.

## Figures and Tables

**Figure 1 diagnostics-13-00659-f001:**
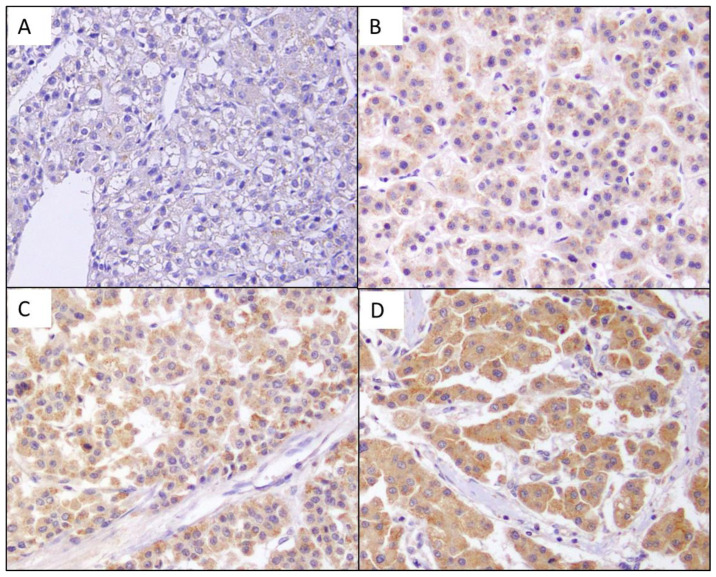
PAD2 was immunohistochemically stained, and its intensity was graded as none (**A**), + (**B**), ++ (**C**) and +++ (**D**). Original magnification × 200.

**Figure 2 diagnostics-13-00659-f002:**
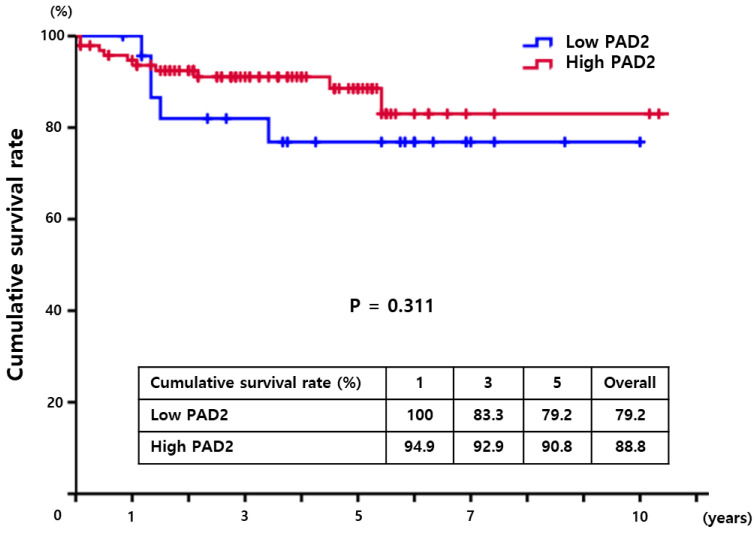
The cumulative survival rates in HCC patients with high PAD2 were higher than those of patients with low PAD (*p* = 0.311). The patients with high PrP^C^ showed cumulative survival rates of 94.9%, 92.9%, and 90.8% at 1, 3, and 5 years, respectively. However, the proportion of patients with low PrP^C^ were 100%, 83.3%, and 79.2%, respectively.

**Figure 3 diagnostics-13-00659-f003:**
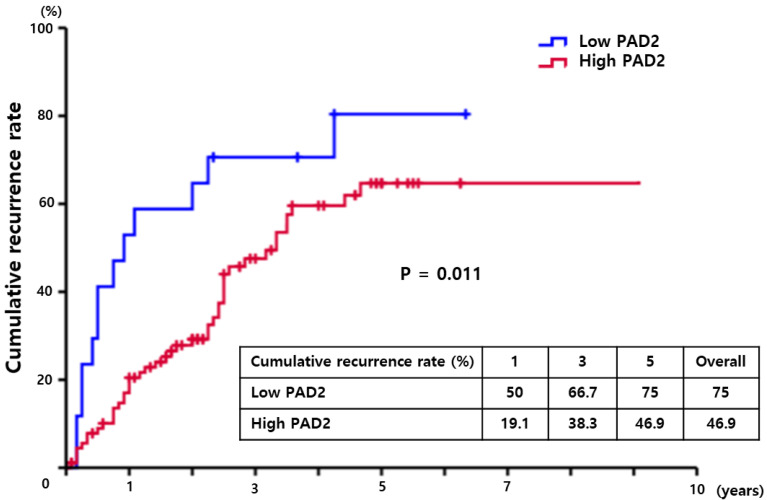
The cumulative recurrence rates in HCC patients with low PAD2 expression were higher than those of patients with high PAD2 expression. The patients with low PAD2 showed cumulative recurrence rates of 50%, 66.7%, and 75.0% at 1, 3, and 5 years, respectively. However, the proportion of patients with high PAD was 19.1%, 38.3%, and 46.9%, respectively (*p* = 0.011).

**Table 1 diagnostics-13-00659-t001:** Clinical characteristics according to PAD2 expression.

Variables	Total (n = 122)	Low (n = 24)	High (n = 98)	*p* Value
Age (years)	62.74 ± 11.876	53.67 ± 11.82	64.96 ± 10.84	<0.001
Sex (male), n (%)	104 (85.2)	18 (75)	86 (87.8)	0.121
HBsAg-positive (+), n (%)	81 (66.4)	21 (87.5)	60 (61.2)	0.016
DM, n (%)	33 (27.0)	3 (12.5)	30 (30.6)	0.122
HTN, n (%)	58 (47.5)	4 (16.7)	54 (55.1)	0.001
LC, n (%)	67 (54.9)	13 (54.2)	54 (55.1)	0.934
Child-Pugh class (B/A)	10/112	2/22	8/90	0.978
Tumor size (≥5 cm/<5 cm)	34/88	8/16	26/72	0.505
Tumor number	1.24 ± 0.59	1.13 ± 0.34	1.27±0.64	0.298
PVI	3 (2.5)	2 (8.3)	1 (1.0)	0.098
Edmonson-Steiner (1, 2/3, 4)	43/79	7/17	36/62	0.487
WBC	5800 (4575–7500)	5400 (3775–7725)	5850 (4800–7425)	0.441
Platelet	167 (110–201)	152.5 (110.5–195.75)	170.5 (108.75–203)	0.301
Serum ALT (IU/L)	29 (21.75–42.25)	27.50 (21.25–46.25)	29.5 (21.75–42.25)	0.900
Total bilirubin	0.675 (0.558–0.962)	0.7 (0.6–0.983)	0.665 (0.55–0.962)	0.604
Albumin	4.2 (3.775–4.4)	4.2 (4.1–4.375)	4.15 (3.7–4.4)	0.824
PT-INR	1.065 (0.998–1.173)	1.065 (1.013–1.188)	1.065 (0.988–1.170)	0.330
AFP > 400, n (%)	23 (18.9)	9 (37.5)	14 (14.3)	0.009

Data are presented as mean ± standard deviation or median (interquartile range) for quantitative variables and n (%) for qualitative variables. ALT, alanine transaminase; AFP, alpha-fetoprotein; DM, diabetes mellitus; HBsAg, hepatitis B virus surface antigen; HTN, hypertension; LC, liver cirrhosis; PT-INR, prothrombin time-international normalized ratio; PVI, portal vein invasion; WBC, white blood cell.

**Table 2 diagnostics-13-00659-t002:** Univariate and multivariate cox analysis for mortality following surgical resection.

Variables	Univariate Analysis			Multivariate Analysis		
	HR	95% CI	*p* Value	HR	95% CI	*p* Value
Age (years)	0.986	0.945–1.028	0.502			
Sex (male/female)	37.535	0.093–15,164.5	0.236			
DM	2.605	0.584–11.610	0.209			
HTN	0.754	0.273–2.086	0.587			
HBsAg-positive (+)	0.606	0.225–1.631	0.321			
LC	0.684	0.254–1.845	0.453			
Child–Pugh class (B/A)	0.870	0.114–6.620	0.893			
Tumor size (≥5 cm/<5 cm)	4.213	1.488–11.929	0.007	4.213	1.488–11.929	0.007
Tumor number (single vs. multiple)	1.681	0.913–3.097	0.096			
Edmonson-Steiner (1, 2/3, 4)	3.430	0.960–12.264	0.058			
Platelet	1.004	0.996–1.012	0.326			
Serum ALT (IU/L)	1.007	0.996–1.019	0.205			
AFP > 400	2.457	0.838–7.208	0.102			
PAD2 (high/low)	0.608	0.209–1.774	0.362			

ALT, alanine transaminase; AFP, alpha-fetoprotein; CI, confidence interval; DM, diabetes mellitus; HBsAg, hepatitis B surface antigen; HR, hazard ratio; HTN, hypertension; LC, liver cirrhosis; PAD2, peptidyl arginine deiminase 2.

**Table 3 diagnostics-13-00659-t003:** Univariate and multivariate Cox analysis for recurrence following surgical resection.

Variables	Univariate Analysis			Multivariate Analysis		
	OR	95% CI	*p* Value	OR	95% CI	*p* Value
Age (years)	0.989	0.966–1.012	0.349			
Sex (male/female)	0.498	0.256–0.966	0.039	0.500	0.253–0.986	0.045
DM	0.769	0.431–1.373	0.375			
HTN	0.902	0.536–1.520	0.699			
HBsAg-positive (+)	0.944	0.548–1.628	0.837			
LC	0.690	0.410–1.160	0.161			
Child–Pugh class (B/A)	1.697	0.726–3.964	0.222			
Tumor size (≥5 cm/<5 cm)	2.392	1.392–4.111	0.002	2.552	1.430–4.554	0.002
Tumor number	1.263	0.829–1.924	0.278			
Edmonson–Steiner (1, 2/3, 4)	1.096	0.640–1.875	0.739			
Platelet	1.002	0.998–1.006	0.417			
Serum ALT (IU/L)	1.009	0.999–1.017	0.068			
AFP > 400	2.062	1.087–3.909	0.027	1.049	0.498–2.210	0.900
PAD2 (low/high)	2.176	1.169–4.052	0.014	2.007	1.010–4.075	0.048

ALT, alanine transaminase; AFP, alpha-fetoprotein; CI, confidence interval; DM, diabetes mellitus; HBsAg, hepatitis B surface antigen; HR, hazard ratio; HTN, hypertension; LC, liver cirrhosis; PAD2, peptidyl arginine deiminase 2.

## Data Availability

No new data were created or analyzed in this study. Data sharing is not applicable to this article.

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
