# Peer review of "Expression of Peptidyl Arginine Deiminase 2 Is Closely Associated with Recurrence in Patients with Hepatocellular Carcinoma"

_diagnostics, 2023, doi:10.3390/diagnostics13040659_

Round 1

Reviewer 1 Report

This study examines the prognostic value of PAD expression following resection for HCC

The study is interesting and well-written and could have clinical interest in the future

I have the following comments:

Abstract and introduction: no comments. Seems appropriate

Statistical analysis: clearly from table 1 not all data are normally distributed. For these data (platelets, ALT, bilirubin) you must give data as median (interquartile range). Also for these data Mann-Whitney must be used in stead of Student’s t.

Figures 2 and 3: I do not understand the numbers given compared with the graphs and the figure text. E.g., 87/98=88.8% in the graph, 90.8% in the text. Must match. Also, add a 1-, 3-, and 5-year mark in the graph to show where you analyze

Discussion and conclusion: fine

Author Response

Reviewer 1

  1. I have the following comments:

Abstract and introduction: no comments. Seems appropriate

Answer: Thank you for your comment.

  1. Statistical analysis: clearly from table 1 not all data are normally distributed. For these data (platelets, ALT, bilirubin) you must give data as median (interquartile range). Also for these data Mann-Whitney must be used instead of Student’s t.

Answer: We would like to thank for your detailed review. We changed the expression of data (WBC, Platelet, ALT, bilirubin, albumin and PT-INR) with median with interquartile range in the Table 1 as you commented. Also, we added a sentence ‘Also, Mann-Whitney test was conducted for comparisons of variables with non-normal distribution between groups’ in the section of statistical analysis.

  1. Figures 2 and 3: I do not understand the numbers given compared with the graphs and the figure text. E.g., 87/98=88.8% in the graph, 90.8% in the text. Must match. Also, add a 1-, 3-, and 5-year mark in the graph to show where you analyze.

Answer: Thank for your detailed comment. The data shown in the figure2 and 3 indicate overall death and recurrence, and the data for 1, 3, and 5 years are indicated in the text, which is thought to have resulted in midunderstanding. Therefore, the figure 2 and 3 has been modified.

Reviewer 2 Report

The significance of PAD2 expression in HCC using operated samples is demonstrated. It is necessary to be modified several points.

1. From the original clinical background, the group with low PAD2 expression tends to be younger, with HBV-related liver cancer, and with higher AFP levels. Since the group with younger HBV and higher AFP is considered to be more likely to relapse in the first place, this point should be analyzed in conjunction with the background.

2. How did you evaluate PAD2 expression in liver cancers that are nodule-in-nodule HCC or have different degrees of differentiation?

3 As for in Table 1A, the cellular density and differentiation looks different from other samples. It does not seem appropriate to evaluate immunostaining. Table 1 should be shown with specimens of the same degree of differentiation as a higher magnification.

Author Response

Reviewer 2

The significance of PAD2 expression in HCC using operated samples is demonstrated. It is necessary to be modified several points.

  1. From the original clinical background, the group with low PAD2 expression tends to be younger, with HBV-related liver cancer, and with higher AFP levels. Since the group with younger HBV and higher AFP is considered to be more likely to relapse in the first place, this point should be analyzed in conjunction with the background.

Answer: Thank for your detailed comment. According to your comment, we conducted multivariate cox analysis for recurrence including HBV infectivity, age, AFP level and PAD expresssion, however, we could not find statistically significance for mentioned factors.

HBV infectivity (HR 0.970, 95% CI 0.530-1.842, p = 0.970), age (HR 1.009 95% CI 0.984-1.035, p = 0.491), AFP level (HR 1.438, 95% CI 0.698-2.964, p = 0.324), PAD2 expression (HR 1.278 95% CI 0.625-2.615, p = 0.501),

  1. How did you evaluate PAD2 expression in liver cancers that are nodule-in-nodule HCC or have different degrees of differentiation?

Answer: Thank you for your detailed comment. There were no nodule-in-nodule HCC included in our cohort. When constructing the tissue microarray, two cores were sampled for each cases, with each core showing homogenous differentiation. The immunostaining intensity and proportion was evaluated for each core and then average the total score for each cases.

  1. As for in Table 1A, the cellular density and differentiation looks different from other samples. It does not seem appropriate to evaluate immunostaining. Table 1 should be shown with specimens of the same degree of differentiation as a higher magnification.

Answer: Thank you for your comment. We changed the representative figures with x200 magnification.
